# Thirty-day hospital readmission in females with acute heart failure and breast cancer: A retrospective cohort study from national readmission database

**Soumya Kambalapalli**[1], **Nischit Baral**[2], **Timir K. Paul**[3]*, **Prakash Upreti**[4],
**Fahimeh Talaei**[1], **Sarah Ayad**[5], **Mahmoud Ibrahim**[1], **Vikas Aggarwal**[6], **Gautam Kumar**[7],
**Chadi Alraies**[8], **Joshua Mitchell**[9]

1 Department of Internal Medicine, McLaren Flint/Michigan State University College of Human Medicine, Flint, MI, United States of America, 2 Department of Cardiology, Mount Sinai Medical Center Miami, Columbia University, Miami, Florida, United States of America, 3 Department of Cardiovascular Medicine, Ascension Saint Thomas Hospital, University of Tennessee College of Medicine, Nashville, Tennessee, United States of America, 4 Rochester General Hospital, Rochester, NY, United States of America, 5 Department of Internal Medicine, Hurley Medical Center/Michigan State University College of Human Medicine, Flint, MI, United States of America, 6 Division of Cardiology, University of Michigan, Ann Arbor, MI, United States of America, 7 Division of Cardiology, Emory University School of Medicine, Atlanta, GA, United States of America, 8 Division of Cardiology, Wayne State University/Detroit Medical Center, Detroit, MI, United States of America, 9 Division of Cardiology, Washington University, St. Louis, MO, United States of America

* timirpaul@gmail.com

## Abstract

### Background

Breast Cancer and cardiovascular diseases are amongst the two leading causes of mortality in the United States, and the two conditions are connected in part because of recognized cardiotoxicity of cancer treatments. The aim of this study is to investigate the predictors risk factors for thirty-day readmission in female breast cancer survivors presenting with acute heart failure.

### Methods

This is a retrospective cohort study of acute heart failure (AHF) hospitalization in female patients with breast cancer in 2019 using the National Readmission Database (NRD), which is the largest publicly available all-payer inpatient readmission database in the United States. Our study sample included adult female patients aged 18 years and older. The primary outcome of interest was the rate of 30-day readmission.

### Results

In 2019, there were 8332 total index admissions for AHF in females with breast cancer and 7776 patients were discharged alive. The mean age was 74.4 years (95% CI: 74, 74.7). The percentage of readmission at 30 days among those discharged alive was 21.8% (n = 1699). Hypertensive heart disease with chronic kidney disease accounted for the majority of

**Data Availability Statement:** Our study's data is available on the public data available on National Readmission Database (NRD), which is the largest

publicly available all-payer inpatient readmission database in the United States. The NRD from the year 2019 which contains discharge data for 58.7% of all US hospitalizations was used. More information about the NRD can be obtained from the website https://hcup-us.ahrq.gov. The study was exempt from the local Institutional Review Board as NRD is a publicly available database with de-identified data sets. Others can access these datasets by logging in to HCUP (https://hcup-us. ahrq.gov/tech_assist/centdist.jsp) and creating an account with an email so that they can access this data in the same manner as the authors. HCUP. HCUP Nationwide Readmissions Database (NRD). Healthcare Cost and Utilization Project (HCUP). 2014, 2016, and 2017. 2023 [cited 2023; Agency for Healthcare Research and Quality, Rockville, MD. www.hcup-us.ahrq.gov/]. Available from: www.hcup-us.ahrq.gov/nrdoverview.jsp.

**Funding:** The author(s) received no specific funding for this work.

**Competing interests:** The authors have declared that no competing interests exist.

readmission in AHF with breast cancer followed by sepsis, acute kidney injury, respiratory failure, pneumonia, and atrial fibrillation. Demographic factors including higher burden of comorbidities predict readmission. The total in-hospital mortality in index admission was 6.67% (n = 556) and for readmitted patients was 8.77% (n = 149). The mean length of stay for index admission was 7.5 days (95% CI: 7.25, 7.75).

## Conclusions

Readmission of female breast cancer survivors presenting with AHF is common and largely be attributed to high burden of comorbidities including hypertension, and chronic kidney disease. A focus on close outpatient follow-up will be beneficial in lowering readmissions.

## Introduction

Cardiovascular diseases and cancer continue to be the two major causes of death in the United States, according to data from 2020 [1]. Breast cancer (BC) is the most prevalent cancer in women, with one in eight women predicted to develop the disease over their lifetime [2]. According to data from the Center for Disease Control, approximately 264,000 women in the United States are diagnosed with breast cancer each year [3]. While early detection of breast cancer through screening and recent developments in therapy have significantly improved survival, cardiovascular death is still high in this population [4]. There is an increasing recognition that the two diseases are intertwined on many levels partly due to the shared risk factors such as obesity, age, diabetes, hypertension, heightened inflammation and the known cardiotoxicity of cancer therapy including chemotherapy and radiotherapy [5]. Breast cancer shares risk factors and common pathophysiological mechanisms with the development of cardiovascular disease. The complications of cancer related treatments along with these risk factors (advancing age and preexisting comorbidities) result in morbidity and decline in quality of life.

In a study by Thavendiranathan et.al, cardiovascular risk in breast cancer was reported as 4.1% chance of cardiovascular event occurrence within 5-year timeframe. But when important risk factors such as advancing age and preexisting cardiovascular conditions are taken into consideration the mortality risk amplifies to about 8.9% withing 5-year timeframe. Overall, the mortality risk increases 3.8-fold in individuals with breast cancer who subsequently develop cardiovascular disease [4]. Thus, given the increasing prevalence of cardiovascular diseases among breast cancer patients, both during and post-treatment, it is imperative to assess the contributing risk factors and develop preventive interventions.

Common neoadjuvant and adjuvant treatments for breast cancer have been linked to a higher risk of cardiac conditions, such as heart failure (HF), arrhythmias, and ischemic heart disease [6]. This highlights the importance of these competing risk factors in breast cancer patients. As per the American Society of Clinical Oncology, individuals with breast cancer exhibit a heightened occurrence of cardiovascular disease (CVD) in comparison to non-cancer controls. Furthermore, after the onset of CVD, overall survival outcomes demonstrate significant deterioration [4]. In patients with breast cancer, CVD contributes to 16.3% of deaths, surpassing mortality from BC in those with pre-existing cardiovascular risk factors at a 10-year follow-up [1, 4]. BC patients exposed to cardiotoxic therapies, such as anthracycline-based chemotherapy and trastuzumab, face an elevated risk of CVD, including HF [7, 8]. Approximately one in four older BC patients succumbs to CVD [9]. Consequently, as the number of

HF cases with a history of BC is expected to rise, understanding the impact of BC on HF survival and treatment becomes increasingly vital.

Heart failure is one of the leading causes of hospitalization and readmission and is major contributor to the growing healthcare burden [10]. Almost one- in- every- four HF patients is readmitted within 30 days of discharge, and roughly half are readmitted within 6 months [11]. Therefore, we through this study we intend to assess the predictors for the thirty-day readmission in female breast cancer survivors presenting with acute heart failure (AHF).

## Methods

We used the National Readmission Database (NRD), which is the largest publicly available all-payer inpatient readmission database in the United States. The Agency for Healthcare Research and Quality (AHRQ), Healthcare Cost and Utilization Project (HCUP) is the official sponsor of the data [12]. The NRD from the year 2019 which contains discharge data for 58.7% of all US hospitalizations was used. More information about the NRD can be obtained from the website https://hcup-us.ahrq.gov. The study was exempt from the local Institutional Review Board as NRD is a publicly available database with de-identified data sets [12].

### Study design and population

This is a retrospective cohort study utilizing the hospitalizations of adult females (≥18 years old) with AHF as the index or principal diagnosis and breast cancer as secondary diagnosis Principal diagnosis is the main diagnosis for which the patients are admitted in the hospital, and it is coded by variable I10_DX1. Secondary diagnosis is the diagnosis either present during the hospitalization or a comorbidity present from before coded by variable I10_DX2 to I10_DX40. To include breast cancer patients who might still be under treatment, we excluded hospitalizations with history of breast cancer. The diagnostic codes for AHF were based on the International Classification of Diseases, Tenth Revision, Clinical Modification (ICD-10-CM) (S1 Table). We excluded December hospitalization as per the NRD data policy as they don't have complete information to assess 30-day readmission along with elective and traumatic hospitalizations.

### Outcome measures

The index admission included initial principal hospitalizations for AHF in patients with breast cancer. The primary outcome of interest was the rate of 30- day readmission rate. Secondary outcomes include predictors of 30-day readmission, top ten principal diagnoses in 30-day readmission, all cause in-hospital mortality rate, and length of stay (LOS) in days. NRD includes a dichotomous variable that identifies whether a patient died in a hospital or not (NIS variable DIED), as well as a continuous variable that represents LOS (NIS variable LOS). The 30-day readmission rate was calculated as the total number of index hospitalizations with AHF and breast cancer who were readmitted for any cause (excluding traumatic or elective admissions) after initial index AHF admissions within 30-days after being discharged alive as numerator divided by the total number of AHF and breast cancer hospitalizations who were discharged alive after index admission [12].

### Statistical analysis

We evaluated the data for the outliers and tested the distribution of the main outcome (30- day readmission rate, in-hospital mortality, and LOS). Mean, median, interquartile range, frequencies, and percentages were calculated to describe the characteristics of the study sample. Chi-$^2$

test was used for categorical variables and results were displayed in proportions. A t-test was used for continuous variables. Univariable cox regression was used to identify variables for multivariable cox regression. Variables with p values less than 0.20 were included in final multivariable cox regression to calculate the hazard ratio for 30-day readmission with a p value less than 0.05 as the level of statistical significance. STATA 17.0 (Stata-Corp., College Station, Tx, USA) was employed for all analysis.

## Results

The total index admissions for AHF in females with breast cancer were 8332; among them, 7776 patients were discharged alive. Mean age was 74.4 years (95% CI: 74, 74.7). The percentage of readmission at 30 days among those discharged alive was 21.8% (n = 1699). The total in-hospital mortality in index admission was 6.67% (n = 556) and for readmitted patients was 8.77% (n = 149). The mean LOS for index admission was 7.5 days (95% CI: 7.25, 7.75). Demographic characteristics of patients with index admission are shown in Table 1.

This study revealed that hypertensive heart disease with chronic kidney disease was the leading cause of readmission in AHF and breast cancer hospitalizations, followed by sepsis, acute kidney injury, respiratory failure, pneumonia, and atrial fibrillation as shown in Table 2.

Readmission for AHF with breast cancer carried more than 34% increase in mortality as shown in Table 3.

Age, comorbidity, teaching status, income, insurance, obesity, acute kidney injury, atrial fibrillation, valvular heart disease, hypertension, acute exacerbation of chronic obstructive pulmonary disease (COPD) and major bleed were included in univariable cox regression. Being on Medicaid, hypertension, higher comorbidities, acute exacerbation of COPD, and anemia were shown to increase readmission in univariable cox regression. Higher age was associated with lower readmission rate.

## Discussion

Our study demonstrated d that hypertensive heart disease with chronic kidney disease was the leading cause of hospital readmission in female breast cancer patients with AHF. Our findings also suggest that patients who were readmitted within 30daysof initial discharge had 8.8-fold higher in-hospital mortality rate than the rate at the initial admission. A higher burden of comorbidities and acute exacerbation of COPD were significant predictors of 30-days readmission. With increase in cancer survivor's secondary to early detection and varied treatment strategies, aging population with chronic comorbidities are a risk for developing cardiovascular conditions leading to decline in functional status, significant morbidity, and impaired health related quality of life.

Few studies have investigated a correlation between the likelihood of developing HF in breast cancer in low socioeconomic status patients. In a study by Sterling et al. involving 690 HF patients across 440 U.S. hospitals, 23.5% exhibited low educational attainment, 63.0% had limited income, 21.0% experienced zip code-level poverty, 43.5% inhabited areas with shortage of Health Professional Areas (HPSAs), 39.3% lived in states with deficient public health infrastructure, 13.1% experienced social isolation, 13.3% had impoverished social networks, and 10.2% resided in rural areas [13]. This study also reported that study participants who were readmitted within 30days had more comorbidities and longer length of stay. Summarizing that patients enrolled in Medicare, a government-funded health insurance program in the United States, have elevated readmission rates and morbidity [10]. As mentioned in the REGARDS study (Reasons for Geographic and Racial Differences in Stroke) [13], these results may be attributable to low health literacy/awareness, higher financial burden associated with

**Table 1. Baseline characteristics of the index admission and readmission in acute heart failure in breast cancer patients.**

| Categories | Variables | Index admission, (%) 8332 | 30-day Readmission n (%)1699 |
|---|---|---|---|
| Age (years) (95% CI) | - | 74.4 (74, 74.7) | 72.9 (72.2, 73.7) |
| Insurance | Medicare | 80.8 | 79.9 |
|  | Medicaid | 6.2 | 7.1 |
|  | Private | 11.1 | 10.9 |
|  | Self-paying | 0.7 | 0.5 |
| Setting/location | Rural | 18.4 | 19.7 |
|  | Urban non-teaching | 73.2 | 72.7 |
|  | Urban teaching | 8.4 | 7.6 |
| Bed size | Small | 21.0 | 21.6 |
|  | Medium | 28.2 | 28.3 |
|  | Large | 50.8 | 50.1 |
| Charlson comorbidity index | 1 | 0.1 | 4.5 |
|  | 2 | 0.1 | 5.1 |
|  | 3 or higher | 99.8 | 89.9 |
| Annual income (US $ per year) | 1–45,999 | 28 | 29.1 |
|  | 46K–58,999 | 26.3 | 27.3 |
|  | 59K-78,999 | 24.9 | 22.4 |
|  | 79K or more | 20.9 | 21.3 |
| Major bleed | - | 1.7 | 3.3 |
| Comorbidities | Obesity | 22 | 22.6 |
|  | Hypertension | 45.6 | 61.9 |
|  | Tobacco Smoker | 24.7 | 24.6 |
|  | Anemia | 42.5 | 45.1 |
|  | Valvular heart disease | 15.8 | 13.5 |
|  | Acute kidney injury | 32.2 | 34.7 |
|  | COPD | 26.9 | 29.4 |
|  | Diabetes mellitus | 43.3 | 47.1 |

SD: Standard Deviation LL: Lower Limit UL: Upper Limit CI: Confidence Interval

COPD: Chronic Obstructive Pulmonary Disease

the cost of cancer treatment, decreased likelihood of adhering to follow-up care, medication non-compliance, and a lack of receiving referrals for specialized medical care based upon insurance coverage [13, 14]. A positive linear correlation exists between the number of socio-economic risk factors and the incidence of hospitalization for HF, cardiovascular events, and mortality, such that an increase in the number of socioeconomic risk factors is associated with an increased risk of these adverse health outcomes [14].

The mortality risk for patients hospitalized for AHF is highest in the early post discharge period, and post discharge hospital events are high in females with breast cancer who are hospitalized for AHF. Previous studies on metastatic cancer or solid tumors without metastasis did not recognize either as an independent predictor of AHF readmission, although in our study the 30-day mortality rate of 8.77% in females with breast cancer was higher than the 30-day mortality rate (4.2%) in the general population, as shown in previous observational studies [14].

Similarly, the 30-day readmission rate for AHF in female breast cancer patients was 21.8%, which is higher than the (18.2%) readmission rate of the general patient population who were

**Table 2. Top 10 principal diagnoses of readmissions for acute heart failure hospitalizations with breast cancer.**

| Principal readmission diagnosis (n = 1695) | Number |
|---|---|
| Hypertensive heart disease with CKD stages 1–4 | 233 |
| Hypertensive heart disease with heart failure | 166 |
| Sepsis, unspecified organism | 136 |
| Acute kidney injury | 62 |
| Acute and chronic respiratory failure with hypoxia | 53 |
| Pneumonia, unspecified organism | 37 |
| Hypertensive heart disease with End stage kidney disease | 34 |
| Acute on chronic diastolic heart failure | 30 |
| Acute respiratory failure with hypoxia | 29 |
| Paroxysmal Atrial fibrillation | 27 |
| **Principal readmission diagnosis (n = 1695)** | **Number (%)** |
| Cardiovascular | 490 (28.9) |
| • Hypertensive heart disease with CKD stages 1–4 | 233 (13.7) |
| • Hypertensive heart disease with *heart failure* | 166 (9.8) |
| • Hypertensive heart disease with End stage kidney disease | 34 (2.0) |
| • Acute on chronic diastolic *heart failure* | 30 (1.8) |
| • Paroxysmal Atrial fibrillation | 27 (1.6) |
| Sepsis, unspecified organism | 136 (8.0) |
| Respiratory | 119 (7.0) |
| • Acute and (on) chronic respiratory failure with hypoxia | 53 (3.1) |
| • Pneumonia, unspecified organism | 37 (2.2) |
| • Acute respiratory failure with hypoxia | 29 (1.7) |
| Acute kidney injury | 62 (3.6) |

CKD: chronic kidney disease

hospitalized for HF in previous NRD studies [14]. Moreover, in our study, patients with a higher Charlson comorbidity index, i.e., a higher burden of comorbidities, were more likely to be readmitted within 30-days following initial hospitalization (HR 1.07; 95% CI 1.04–1.10; $p<$ 0.001) as seen in Table 4. This is similar to previous data showing that Charlson comorbidity index is a predictor of 30-day readmission in heart failure patients [15–19].

In this real-world study of a selected group of females with breast cancer, the leading diagnosis of readmission was cardiovascular causes (28.9%). In the general population, it is 49.7% [20], followed by sepsis (8%), respiratory causes (7%), and acute kidney injury (3.6%). Among cardiovascular causes of AHF readmission, the most common causes of principal readmission were hypertensive heart disease with CKD (13.7%) and hypertensive heart disease with HF (9.8%). In other NRD studies in the general patient population, cardiovascular causes were also leading causes of readmission, although percentage was higher (52.8%), with HF being the cause of one-third of readmissions [17].

**Table 3. Clinical outcomes of acute heart failure hospitalizations with breast cancer.**

| Outcome | 30-day readmission, % | Index admission, % | OR/MD (95% CI) | P value |
|---|---|---|---|---|
| In hospital mortality | 8.77 (n = 149) | 6.67 (n = 556) | 1.34 (1.11, 1.62) | 0.002 |
| Length of stay | 6.4 days | 7.5 days | -1.10 (-1.52, -0.68) | <0.001 |

OR: Odds ratio, MD: Mean difference

**Table 4. Unadjusted and adjusted hazard ratios for predictors of readmission among acute heart failure with breast cancer patients.**

| 30-day Readmission Predictors | Unadjusted HR | LL, UL 95% CI | P value | Adjusted HR | LL, UL 95% CI | P value |
|---|---|---|---|---|---|---|
| Age | 0.99 | 0.98, 0.99 | 0.001 | 0.99 | 0.98, 0.99 | <0.001 |
| Charlson Comorbidity index | 1.08 | 1.06, 1.11 | <0.01 | 1.07 | 1.04, 1.10 | <0.001 |
| Setting/teaching status of Hospital (Reference: Rural) | | | | | | |
| Non-teaching | 0.93 | 0.79, 1.10 | 0.412 | 0.92 | 0.78, 1.08 | 0.318 |
| Teaching | 0.76 | 0.56, 1.02 | 0.071 | 0.74 | 0.54, 1.02 | 0.064 |
| Household Income by Quartile (USD) (Reference: 1–45,999) | | | | | | |
| 46,000–58,999 | 0.98 | 0.81, 1.19 | 0.852 | 1.01 | 0.84, 1.22 | 0.925 |
| 59,000–78,999 | 0.87 | 0.72, 1.05 | 0.135 | 0.88 | 0.73, 1.06 | 0.194 |
| 79,000 or higher | 0.98 | 0.82, 1.19 | 0.874 | 1.04 | 0.86, 1.25 | 0.700 |
| Insurance type (Reference: Medicare) | | | | | | |
| Medicaid | 1.39 | 1.08, 1.79 | 0.011 | 1.08 | 0.82, 1.41 | 0.602 |
| Private | 1.07 | 0.85, 1.35 | 0.572 | 0.90 | 0.70, 1.18 | 0.452 |
| Obesity | 1.13 | 0.97, 1.32 | 0.118 | 1.04 | 0.88, 1.22 | 0.602 |
| Acute kidney injury | 1.22 | 1.06, 1.40 | 0.006 | 1.12 | 0.97, 1.30 | 0.128 |
| Atrial Fibrillation | 0.88 | 0.77, 1.01 | 0.069 | 0.97 | 0.84, 1.12 | 0.701 |
| Anemia | 1.20 | 1.05, 1.37 | 0.008 | 1.08 | 0.94, 1.24 | 0.252 |
| Valvular Heart Disease | 0.97 | 0.81, 1.16 | 0.736 | NA | NA | NA |
| AECOPD | 1.25 | 1.08, 1.44 | 0.002 | 1.22 | 1.05, 1.41 | 0.010 |
| Major Bleed | 0.84 | 0.49, 1.45 | 0.528 | NA | NA | NA |
| Hypertension | 1.14 | 1.00, 1.30 | 0.049 | 1.12 | 0.98, 1.28 | 0.095 |

AECOPD: acute exacerbation of chronic obstructive pulmonary disease

LL–lower limit; UL–upper limit

Data from clinical trials had previously identified admission serum creatinine, low systolic blood pressure, and pulmonary disease as the important predictors for the combined end point of death or rehospitalization [20]. Also, kidney disease, COPD, diabetes mellitus and female sex were major independent risk factors for readmission [19], but not a diagnosis of solid tumor or metastatic cancer. In the current study, the most frequent variables associated with increased HF readmission in females with breast cancer was being on Medicaid, hypertension, higher comorbidities, acute exacerbation of COPD, and anemia, but older age was associated with a lower readmission rate, which is consistent with similar reports from Medicare beneficiaries and proposed predictive models of HF rehospitalization [16, 19, 21, 22].

Our study has several limitations. First, this is an observational analysis of NRD administrative data, thus the result cannot establish causality. Second, data regarding type of HF, stage of cancer, previous or current chemotherapeutic regimen, and post-discharge medications, clinic, or specialist follow-up were not available, which may have impacted readmission mortality rates after index hospitalization. Third, NRD excludes interstate hospitalizations and does not link patient data across years, which may have affected readmission estimates. Fourth, the use of ICD codes for administrative data analysis is subject to potential errors related to coding discrepancies, as the accuracy of the data is dependent on the rigor of the coding practices of participating institutes. Finally, NRD only collects data on in-hospital mortality; outpatient death data are not available. Thus, although studies have evaluated the association of hospital readmission reduction programs with post-discharge mortality, such an assessment was not possible in the current study.

## Conclusion

Compared to the general population, the 30-day readmission rate for AHF was significantly higher among female breast cancer patients. Additionally, the in-hospital mortality rate of female breast cancer patients readmitted for AHF within 30 days of their initial discharge was higher than that of the general population. Sepsis, respiratory causes, and acute kidney injury were the most prevalent predictors associated with readmission rates (per Table 4 above). A focus on close outpatient follow-up will be beneficial in lowering readmissions. The results of the present study underscore the significance of implementing strategies aimed at enhancing cardiovascular health among breast cancer patients to prevent morbidity and mortality.

## Supporting information

**S1 Table.**
(DOCX)

## Author Contributions

**Methodology:** Nischit Baral.

**Resources:** Sarah Ayad, Mahmoud Ibrahim.

**Writing – original draft:** Soumya Kambalapalli, Fahimeh Talaei.

**Writing – review & editing:** Timir K. Paul, Prakash Upreti, Vikas Aggarwal, Gautam Kumar, Chadi Alraies, Joshua Mitchell.

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
