## [Decision Letter · Decision Letter 0]

14 Dec 2023

PONE-D-23-24874Thirty-day Hospital Readmission in Females with Acute Heart Failure and Breast Cancer: A Retrospective Cohort Study from National Readmission DatabasePLOS ONE

Dear Dr. Kambalapalli,

Thank you for submitting your manuscript to PLOS ONE. After careful consideration, we feel that it has merit but does not fully meet PLOS ONE’s publication criteria as it currently stands. Therefore, we invite you to submit a revised version of the manuscript that addresses the points raised during the review process.

We look forward to receiving your revised manuscript.

Kind regards,

Chiara Lazzeri

Academic Editor

PLOS ONE

5. Please include your tables as part of your main manuscript and remove the individual files. Please note that supplementary tables (should remain/ be uploaded) as separate ""supporting information"" files"

6. Please include a copy of Table S1 in supporting information file which you refer to in your text on page 13.

Reviewers' comments:

Reviewer's Responses to Questions

**Comments to the Author**

1. Is the manuscript technically sound, and do the data support the conclusions?

Reviewer #1: Yes

Reviewer #2: Yes

2. Has the statistical analysis been performed appropriately and rigorously? 

Reviewer #1: Yes

Reviewer #2: Yes

3. Have the authors made all data underlying the findings in their manuscript fully available?

Reviewer #1: Yes

Reviewer #2: Yes

4. Is the manuscript presented in an intelligible fashion and written in standard English?

Reviewer #1: Yes

Reviewer #2: Yes

5. Review Comments to the Author

Reviewer #1: Introduction: Would add more info on prevalence of HF in Breast cancer patients? what cardiovascular causes are the major cause of death in Breast cancer patients- like CHF/MI etc. This would explain why tis study is important

- Almost-one-in =four (this is hw it appeared on manuscript for review). please check?

Study design:

- If AHF was the principal diagnosis - does it mean just ICD codes for acute heart failure(I50)? If thats the case then what about hypertensive heart and chronic kidney disease with heart failure, code I13. This diagnosis is more often the principal in pts with HF and ckd. Was the ICD code for this used?

- Need to update what icd10 codes were used (for heart failure)

- So were patients with history of breast cancer who were in remission excluded for this study purpose?

Results:

- hypertensive heart disease with chronic kidney disease seems to be the major cause of readmission. As mentioned about hypertensive heart disease with chronic kidney disease with heart failure(I30) is the principal diagnosis in most patients who have CKD as well so it's important to know if this ICD10 was used for extracting data in first place?

Discussion

- first para seems repeat of what was already said above in results

- second para needs to be modified - seems widely jumping from one to other- initially talk about Several studies have investigated a robust correlation between the likelihood of developing HF and breast cancer". Then at end of the para states "underlying association between hypertension and the risk of breast cancer has not been extensively researched or studied yet". This study if apt CHF and breast cancer so narrow down to that- pathophysiology, what evidence other studies show. Also talks about medicare pts , REGARD study which is more about stroke rather than HF. Please use data about Medicare patients and HF readmission if needed.

Reviewer #2: ncrease readmission rate as expected on the lines of other reasearch studies/ literature we already have

if you can try to add other variables if available would be be more novel.

overall good writing and research methodolgy used.

6. PLOS authors have the option to publish the peer review history of their article (what does this mean?). If published, this will include your full peer review and any attached files.

Reviewer #1: No

Reviewer #2: **Yes: **Thulasi Ram Gudi

---

## [Author Response · Author response to Decision Letter 0]

9 Mar 2024

Dear Editor-in-Chief and Reviewers,

We submit for the revision of our manuscript entitled " Thirty-day Hospital Readmission in Females with Acute Heart Failure and Breast Cancer: A Retrospective Cohort Study from National Readmission Database” for consideration in your respected journal. 

We are grateful for your critical comments and thoughtful suggestions which are valuable and very helpful for revising and improving our paper. We truly appreciate your time, effort, and patience towards providing these queries and suggestions.

The following is a point-by-point response to the reviewers’ comments.

Reviewer #1:

Question: Introduction: Would add more info on prevalence of HF in Breast cancer patients? what cardiovascular causes are the major cause of death in Breast cancer patients- like CHF/MI etc. This would explain why tis study is important

Almost-one-in =four (this is hw it appeared on manuscript for review). please check?

Response: Thank you for the query and suggestions. 

We have reviewed literature on HF prevalence in Breast cancer and added it to the introduction as suggested. Also made appropriate changes regarding the incidence of cardiovascular complications in breast cancer patients. 

Reviewer Question: 

Study design:

- If AHF was the principal diagnosis - does it mean just ICD codes for acute heart failure(I50)? If thats the case then what about hypertensive heart and chronic kidney disease with heart failure, code I13. This diagnosis is more often the principal in pts with HF and ckd. Was the ICD code for this used?

- Need to update what icd10 codes were used (for heart failure)

- So were patients with history of breast cancer who were in remission excluded for this study purpose?

Response: Thank you for the query and suggestions. 

Following ICD codes were used for Acute heart failure, Hypertensive heart disease and chronic kidney disease for our study’s analysis. 

Below is the table depicting the ICD codes used for analysis. 

CONDITION ICD CODE

Acute Heart Failure I50.21, I50.23, I50.31, I50.33, I50.41, I50.43

Hypertension I10

Hypertensive Chronic Kidney disease I12.9, I12.0

Hypertensive heart and chronic kidney disease with heart failure, with stage 1 through stage 4 chronic kidney disease, or unspecified chronic kidney disease I13.0

Hypertensive heart and chronic kidney disease without heart failure I13.1

Hypertensive heart and chronic kidney disease without heart failure, with stage 5 chronic kidney disease, or end stage renal disease I13.11, I13.2

Renovascular hypertension I15.0

Other secondary hypertension I15.8, I6.74

Hypertensive heart and chronic kidney disease without heart failure I13.1

Hypertensive heart and chronic kidney disease without heart failure, with stage 5 chronic kidney disease, or end stage renal disease I13.11, I13.2

Renovascular hypertension I15.0

Other secondary hypertension I15.8, I6.74

Reviewer Question: So, were patients with history of breast cancer who were in remission excluded for this study purpose? 

Response: Thank you for the query. 

All patients with history or active breast cancer were included in the study with following ICD codes. 

Breast cancer ICD codes- 

C50.011, C50.012, C50.019, C50021, C50022, C50029, C50111, C50112, C50119, C50121, C50122, C50129, C50211, C50212, C50219, C50221, C50222, C50229, C50311, C50312, C50319, C50321, C50322, C50329, C5041, C50412, C50419, C50421, C50422, C50429, C50511, C50512, C50519, C50521, C50522, C50529, C50611, C50612, C50619, C50621, C50622, C50629, C50811, C50812, C50819, C50821, C50822, C50829, C50911, C50912, C50919, C50921, C50922, C50929

Reviewer Question: hypertensive heart disease with chronic kidney disease seems to be the major cause of readmission. As mentioned about hypertensive heart disease with chronic kidney disease with heart failure(I30) is the principal diagnosis in most patients who have CKD as well so it's important to know if this ICD10 was used for extracting data in first place?

Response: Thank you for the query and suggestions. 

Yes, we have included both hypertensive heart disease and chronic kidney diseases ICD codes for study’s data extraction.

Hypertensive Chronic Kidney disease- I12.9, I12.0

Hypertensive heart and chronic kidney disease with heart failure (with stage 1 through stage 4 chronic kidney disease, or unspecified chronic kidney disease)-I13.0

Hypertensive heart and chronic kidney disease without heart failure- I13.1

Hypertensive heart and chronic kidney disease without heart failure (with stage 5 chronic kidney disease, or end stage renal disease)- I13.11, I13.2

Reviewer Question: 

Discussion

- first para seems repeat of what was already said above in results

- second para needs to be modified - seems widely jumping from one to other- initially talk about Several studies have investigated a robust correlation between the likelihood of developing HF and breast cancer". Then at end of the para states "underlying association between hypertension and the risk of breast cancer has not been extensively researched or studied yet". This study if apt CHF and breast cancer so narrow down to that- pathophysiology, what evidence other studies show. Also talks about medicare pts, REGARD study which is more about stroke rather than HF. Please use data about Medicare patients and HF readmission if needed.

Response: Thank you for the suggestions. 

We have revised the first paragraph of the discussion as advised. We have also reviewed existing literature and made changes in the second paragraph to address their findings correlating with our analysis. We included the findings of REGARDS study as it reported the impact of socioeconomic parameters, social determinants of life (poverty, insurance type, education, household income) in medicare beneficiaries on the long-term outcomes, morbidity and rehospitalization rates in HF patients.

REGARDS STUDY by Sterling, et.al

Sterling MR, Ringel JB, Pinheiro LC, et al. Social Determinants of Health and 30-Day Readmissions Among Adults Hospitalized for Heart Failure in the REGARDS Study. Circ Heart Fail. 2022;15(1):e008409.

Reviewer #2: increase readmission rate as expected on the lines of other research studies/ literature we already have

if you can try to add other variables if available would be more novel.

overall good writing and research methodology used.

Response: Thank you for your kind words and suggestions. 

We aimed at studying readmission rates for heart failure patients in breast cancer using National Readmission Database. Our objective, as a nascent research team, was to investigate the impact of medical comorbidities and socioeconomic parameters in relation to HF readmission rates in breast cancer. Your input is invaluable, and we surely plan to incorporate additional variables into our future studies based on your recommendations.

---

## [Editor Report · Decision Letter 1]

20 Mar 2024

Thirty-day Hospital Readmission in Females with Acute Heart Failure and Breast Cancer: A Retrospective Cohort Study from National Readmission Database

PONE-D-23-24874R1

Dear Dr. Kambalapalli,

We’re pleased to inform you that your manuscript has been judged scientifically suitable for publication and will be formally accepted for publication once it meets all outstanding technical requirements.

An invoice for payment will follow shortly after the formal acceptance. To ensure an efficient process, please log into Editorial Manager at Editorial Manager® , click the 'Update My Information' link at the top of the page, and double check that your user information is up-to-date. If you have any billing related questions, please contact our Author Billing department directly at authorbilling@plos.org.

Kind regards,

Chiara Lazzeri

Academic Editor

PLOS ONE
---

## [Editor Report · Acceptance letter]

13 Jul 2024

PONE-D-23-24874R1 

PLOS ONE

Dear Dr. Kambalapalli, 

I'm pleased to inform you that your manuscript has been deemed suitable for publication in PLOS ONE. Congratulations! Your manuscript is now being handed over to our production team.

Kind regards, 

on behalf of

Dr. Chiara Lazzeri 

Academic Editor

PLOS ONE